# FLEX: END-TO-END TEXT-INSTRUCTED VISUAL NAVIGATION WITH FOUNDATION MODELS

## ABSTRACT

End-to-end learning directly maps sensory inputs to actions, creating highly integrated and efficient policies for complex robotics tasks. However, such models are tricky to efficiently train and often struggle to generalize beyond their training scenarios, limiting adaptability to new environments, tasks, and concepts. In this work, we investigate the minimal data requirements and architectural adaptations necessary to achieve robust closed-loop performance with vision-based control policies under unseen text instructions and visual distribution shifts. To this end, we design datasets with various levels of data representation richness, refine feature extraction protocols by leveraging multi-modal foundation model encoders, and assess the suitability of different policy network heads. Our findings are synthesized in **Flex** (**F**ly-**lex**ically), a framework that uses pre-trained Vision Language Models (VLMs) as frozen patch-wise feature extractors, generating spatially aware embeddings that integrate semantic and visual information. These rich features form the basis for training highly robust downstream policies capable of generalizing across platforms, environments, and text-specified tasks. We demonstrate the effectiveness of this approach on quadrotor fly-to-target tasks, where agents trained via behavior cloning on a small simulated dataset successfully generalize to real-world scenes, handling diverse novel goals and command formulations.

## 1 INTRODUCTION

A significant dimension of human reasoning is mediated through the combination of vision and language, facilitating our mobility in the physical world and our ability to follow directions. Such flexibility and concept understanding are highly desirable in autonomous robots, enabling interactions with humans and handling variants of complex real-world tasks from few representative examples. This inspires an exploration of the conditions necessary to equip robots with a human-like intuition and capacity to execute tasks across various contexts.

Despite advancements in end-to-end deep learning for autonomous navigation, these systems remain largely black-box, lacking interpretability, adaptability, and the ability to generalize far beyond the scope of training data. In contrast, VLMs have demonstrated robust open-world visual understanding across tasks like classification, detection, and segmentation. These models are increasingly adopted in robotics for open-vocabulary detection, object manipulation, and planning, but their reliance on modular pipelines and global embeddings limits their utility for end-to-end robot learning.

To overcome these challenges, we embrace a minimalist design philosophy, leveraging pre-trained vision-language encoders with lightweight adaptations and minimal training data. By extracting fine-grained, text-fused features from patch-level embeddings, our approach bridges the gap between global text and visual understanding and the spatial, context-aware reasoning required for robotics. This streamlined methodology achieves strong generalization on out-of-distribution scenarios while maintaining efficiency, offering a unified framework for vision-based robotic learning tasks.

Hence, we introduce **Flex**, a minimalist methodology that pioneers the integration of a data-efficient approach with open-set capabilities into a robotic framework. We provide a foundational proof-of-concept demonstrating the potential of leveraging VLM features for user-interactive, end-to-end visual navigation agents, offering the flexibility to interpret open-set text instructions at both the object and environment levels. By focusing on basic instructions, we address the core challenges of this novel integration without the added complexity of intricate language processing. This streamlined

approach ensures a thorough understanding of each component and establishes a strong foundation for the framework. Our key contributions are:

- The identification of the core components needed for robust multi-modal generalization in robotic tasks, combining spatial and lexical features via patch-wise descriptors from VLMs.
- The development of a training pipeline for closed-loop visual navigation agents that generalize across unseen environments, using real-time natural language instructions to achieve adaptability well beyond the training scope.
- Extensive experiments on drone fly-to-target tasks, showcasing the ability to generalize from limited simulated training data to diverse real-world scenarios, successfully adapting to new objects, environments, and text instructions.

## 2 PRELIMINARIES

**End-to-end multi-modal imitation learning.** The setup considered is that of an end-to-end control system $f$ that generates commands $\boldsymbol{u} \in \mathbb{R}^n$ where $n$ is the dimension of the output vector. The system takes multi-modal input comprising of a RGB image $\mathsf{I} \in \mathbb{R}^{h \times w \times 3}$, with $h, w$ representing the frame height and width respectively, and a natural language text command $T$. $f$ can be seen as the composition of a feature extraction backbone $\phi$ and a policy head $\pi$, such that $f = \pi \circ \phi$, and yielding control commands through $\boldsymbol{u} = f(\mathsf{I}, T) = \pi(\phi(\mathsf{I}, T))$.

Throughout this work, we do not seek to train or fine-tune $\phi$, but instead, investigate how architectural choices leveraging frozen VLM encoders can yield dense feature representations $\mathsf{F} \in \mathbb{R}^{h' \times w' \times d}$ that integrate both spatial and semantic information tailored to robotics applications (Figure 1). We thus only train the policy network head $\pi$, parameterized by weights $\theta$ adopting the Imitation Learning (IL) paradigm of learning from expert demonstrations.

Indeed, given a dataset $\mathcal{D} = \{(\mathsf{I}_i, T_i, \boldsymbol{u}_i)\}_{i=1}^N$ consisting of $N$ samples, where each sample contains an RGB image $\mathsf{I}_i$, a natural language command $T_i$, and a ground truth control command $\boldsymbol{u}_i \in \mathbb{R}^n$, the policy network $\pi_\theta$ is trained to minimize the Mean Squared Error (MSE) between the predicted control command $\hat{\boldsymbol{u}}_i = \pi_\theta(\phi(\mathsf{I}_i, T_i))$ and the ground truth label $\boldsymbol{u}_i$. With the notation adopted the training objective $\mathcal{L}$ is given in equation 1.

$$\mathcal{L}(\theta) = \frac{1}{N} \sum_{i=1}^N \|\hat{\boldsymbol{u}}_i - \boldsymbol{u}_i\|_2^2 \tag{1}$$

**Autonomous Drone Fly-to-target Task.** The scope of this research extends to a broad array of robotics tasks that rely on the use of both images and text. In the interest of cohesive illustration, we delve into a single running example throughout this manuscript. We explore quadrotor flight and more specifically a vision-based fly-to-target task where the goal can be specified by the human user via natural language. In this context, the control command $\boldsymbol{u} \in \mathbb{R}^4$ comprises of scalar translation velocities $v_x, v_y, v_z$ and the drone's desired yaw rate $\dot{\psi}$. The input RGB frame $\mathsf{F} \in \mathbb{R}^{224 \times 224 \times 3}$ is obtained from the drone's front-facing camera and the text command $T$ is provided by the user.

**Problem statement.** We seek to establish the bare design criteria for training robust, text-instructed, end-to-end control agents. Specifically, our goal is to delineate the conditions for effective leveraging of off-the-shelf models to extract meaningful features suitable for compact downstream policy networks. Our agents should not only excel in learning tasks from very simplified datasets in simulation but also demonstrate robust generalization capabilities to handle previously unseen scenarios. We probe into the three pillars of the IL framework and attempt to answer the following questions:

1. **Dataset design:** What is the minimal degree of data diversity required to obtain sufficiently rich feature representations?
2. **Feature extractor:** What are the suitable feature extractors for text and vision-based robotics learning? How should they be employed to offer potent downstream generalization capability?
3. **Policy network:** What is the impact of the choice of policy network architecture on the performance and interpretability of the trained agents?

## 3 METHODS

### 3.1 TRAINING DATA

A desirable property for an imitation learning system is to master a task from a handful of representative expert demonstrations without requiring extensive enumeration of use cases or intensive randomization and augmentation techniques. Hence, relying on internet-scale trained VLMs for feature extraction largely mitigates the impediments on training dataset size, diversity and augmentation. We investigate the extent to which this statement holds, and limit ourselves to the use of a single simulated scene to generate four training datasets, evaluating the impact of diversity in the goal and text instruction phrasing on generalization capabilities of trained agents:

*1.* One object and one command, containing demonstrations reaching a single goal object (red sphere) with a single command syntax ("Fly to the red ball").

*1M.* One object and multiple commands, with the same goal object, but each run instructed with a lexical alternative of the instruction (as discussed in Appendix A.3).

*2.* Two objects and one command, with red and blue spheres as the example goals and single command wording in either case ("Fly to red/blue ball").

*2M.* Two objects and multiple commands, containing both colored spheres and variations of the syntax between demonstrations. (Training run sequence frames are provided in Figure 10)

### 3.2 PATCH-WISE TEXT-VISION SPATIAL FEATURES

**Generic image-text features.** Robust OoD generalization relies on universal rather than domain-specific features for policy learning. Foundation model encoders leverage internet-scale data to learn generic features from a wide spectrum of contexts. Moreover, incorporating textual instructions demands an extractor capable of seamlessly integrating text inputs with visual features. Thus, a natural choice is pre-trained VLM as our feature extractor cornerstone. More specifically, BLIP-2 (Li et al., 2023) is used throughout this work, as it is specifically designed to fuse multi-modal information from large-scale textual and visual datasets, providing a cohesive representation.

**Spatial resolution for robotic tasks.** Foundation models typically output a global feature vector representing the entire image. This coarse representation is unsuitable for robotics, where policy learning depends on fine-grained spatial information to effectively interpret and respond to the scene. Thus, we propose a method to extract spatial feature vectors for specific areas in an image. To obtain the global descriptor for a frame, we collect such features for multiple areas/patches covering the whole image. Specifically, given an input frame/image $\mathsf{I} \in \mathbb{R}^{h \times w \times 3}$, an input text command $T$, and the patch-resolution $h' \leq h, w' \leq w$, we provide a method which utilizes a multi-modal foundation model $\texttt{VLM} : \mathbb{R}^{h \times w \times 3} \to \mathbb{R}^d$ to derive a tensor of feature descriptors $\mathbf{F} \in \mathbb{R}^{h' \times w' \times d}$, that fuses all the semantic information of $\mathsf{I}$ with the text input $T$ and maintains its location in the scene. For simplicity purposes, we equate $h'$ and $w'$ to the number of (non-overlapping) patches used to divide the input image $\mathsf{I}$ when applying $\texttt{VLM}$ on it (we will discuss how $h'$ and $w'$ can be adjusted to any values $\leq h, w$, respectively) and $n = h'w'$.

**Notations.** Let $\texttt{IMG-DESC}$ be the image encoder of $\texttt{VLM}$ consisting of $L$ layers. For every layer $\ell \in \{1, \cdots, L\}$, we use $Q^\ell, K^\ell \in \mathbb{R}^{n \times d_k}, V^\ell \in \mathbb{R}^{n \times d}$ to denote the output query, key, and value matrices of the $\ell$th attention layer, during the feedforward pass of applying $\texttt{IMG-DESC}$ on $\mathsf{I}$, i.e., during $\texttt{IMG-DESC}(\mathsf{I})$. We now describe our mechanism for extracting patch-text fused features $\mathbf{F}^{(j)}$ for a single patch $\mathsf{I}^{(j)}$, where $j \in \{1, \cdots, n\}$. Then, this can be applied sequentially or in parallel to all patches.

**Single patch feature extraction.** To derive the feature vector $\mathbf{F}^{(j)}$ for the $j$th patch, we introduce an attention mask $m^{(j)} = (m_1^{(j)}, \cdots, m_n^{(j)})^T \in [0, 1]^n$. Each component $m_i^{(j)}$ within this vector, ranging between 0 and 1, determines the contribution of the $i$th patch to the target patch feature $\mathbf{F}^{(j)}$. For instance, to completely exclude patch $i$, set $m_i^{(j)} = 0$. Additionally, to control the masking, we introduce $\alpha \ll 0$ as the parameter controlling the intensity of the masking effect; as $|\alpha|$ increases, the masking effect becomes stronger.

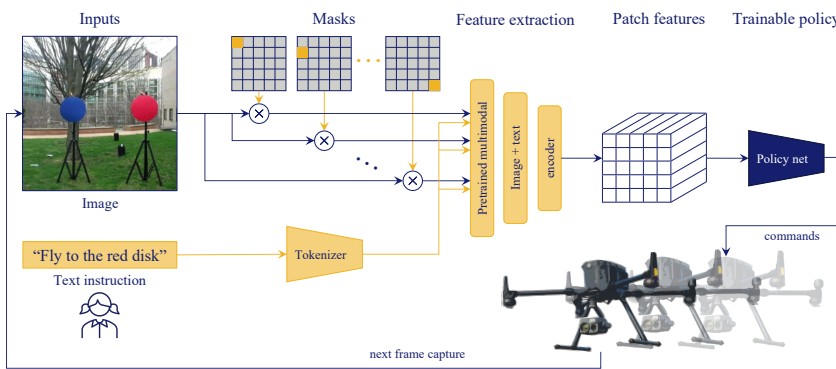

Figure 1: **Flex** pipeline: The drone front view frame capture is successively masked then, in conjunction with a user-specified text instruction, encoded via a pre-trained VLM to create a grid of rich per patch features. A trainable policy network computes the translation velocities and yaw rate commands to be executed by the quadrotor.

Now, to extract the patch feature vector $\mathbf{F}^{(j)}$, we propose to modify the $\ell$-th attention layer to employ the masking provided by $m$ as follows:

1. Set $M^{(j)} = [m^{(j)}, \cdots, m^{(j)}]^T \in \mathbb{R}^{n \times n}$; a matrix of $n$ rows each is equal to $m^{(j)}$, and define $\mathbf{1} \in \mathbb{R}^{n \times n}$ to be an all-ones matrix.

2. Compute $G^\ell := Q^\ell (K^\ell)^T$; the matrix multiplication of the key and query matrices at the $\ell$-th attention layer.

3. A masked version $\hat{G}^{\ell,(j)}$ of $G^\ell$ which focuses on the features of the patches (area) described by $m^{(j)}$ is computed as
$$\hat{G}^{\ell,(j)} = G^\ell + (\mathbf{1} - M^{(j)}) \cdot \alpha,$$
This operation adjusts the attention scores in $\hat{G}^{\ell,(j)}$ according to the mask vector $m^{(j)}$. The $\mathbf{1} - M^{(j)}$ term ensures that patches with an attention mask of 1 remain unchanged, while those with a mask near 0 have their scores reduced to $\alpha$, effectively masking them.

4. With the modified attention scores, the final attention weights are obtained using the softmax function. The attention layer output is now computed as:
$$F_\ell^{(j)} := \text{SoftMax}(\hat{G}^{\ell,(j)})(V^\ell)^T. \tag{2}$$

Notably, we use values of $\alpha \ll 0$ with a very large $|\alpha|$. Observe that at the end of this process, when $m_i^{(j)} = 0$, the corresponding descriptor in $\hat{G}^{\ell,(j)}$ becomes a vector where all entries are approximately $\alpha$. Since $\alpha$ is a very large negative number (e.g., assumably $-\infty$), the result after applying the soft operation will cause its contribution to be close to 0 thus not affecting the final output. When $m_i^{(j)} = 1$, the corresponding descriptor in $\hat{G}^{\ell,(j)}$ is not affected at all, thus, its contribution remains the same through the process.

**Text-Patch fusion.** Let `TEXT-DESC` be the Text Encoder of `VLM`, and let `TEXT-IMG-Fusion` be its text-vision fusion block. Following the $\ell$th attention layer, its output is fed standardly as input to the remaining vision encoder model as `IMG-DESC`$^{\ell \rightarrow}(F_\ell^{(j)})$, where, `IMG-DESC`$^{\ell \rightarrow}$ denotes the remaining part of the vision encoder of the foundation model after the $\ell$-th layer. In parallel to the vision encoding, the text command $T$ is encoded via the text encoder `TEXT-DESC`$(T)$, and then both text and patch descriptors are fused to create the final text-patch fused descriptor as

$$\mathbf{F}^{(j)} := \texttt{TEXT-IMG-Fusion}(\texttt{IMG-DESC}^{\ell \rightarrow}(F_\ell^{(j)}), \texttt{TEXT-DESC}(T)).$$

We note that this method can be extended to any region-wise feature extraction by generalizing the definition of patches to include arbitrarily shaped regions.

**Extracting $m \times m$ resolution descriptors.** Let $h'$ and $w'$ denote the number of non-overlapping patches used to divide the input image $\mathbf{I}$ by `VLM`. For simplicity, we set $h' = w' = 16$ (as in BLIP-2)

and define $m \leq w'$ as the desired resolution, where $w'$ is divisible by $m$. We have $w'^2$ patches ($16 \times 16$ in BLIP-2), each identified by its coordinates $(x, y)$ on the grid ($x, y \in \{1, \cdots w'\}$). While extracting a descriptor for every patch provides detailed information, we seek a minimalist design and smaller resolutions for simpler training. To extract $m \times m$ feature descriptors, we split the $w' \times w'$ grid into $m \times m$ sub-grids, each containing $w'/m \times w'/m$ patches. We extract a descriptor for each sub-grid by setting the corresponding coordinates in the mask vector $m$ to 1 and the rest to 0.

We consider multiple resolutions, splitting the image into 1 (entire image, mask $16 \times 16$), 4 (masks $8 \times 8$), 16 (masks $4 \times 4$), 64 (masks $2 \times 2$), and 256 (masks $1 \times 1$) square patches. Detailed examples of grid splits and masks are provided in Section A.1 of the appendix.

### 3.3 POLICY NETWORK

We aim to identify the most effective policy network architecture for learning from generic extracted features with limited simulated training data. Ideally, we want to preserve text-patch details while aggregating information across layers, enabling decisions from nuanced, context-rich descriptors tailored to the task. Vision Transformers (ViT) are of interest as they maintain spatial resolutions across layers. We also consider simpler architectures such as Convolutional Neural Networks (CNN or Conv), and Multi-Layer Perceptrons (MLP) for learning the control policy. A detailed description of each model and its parameters is provided in Appendix A.5.

## 4 EXPERIMENTS

### 4.1 FLY-TO-ANY-TARGET TASK

**Task description.** The objective is to develop a vision-based quadrotor navigation agent capable of reaching arbitrary user-specified goals present in its field of view (FOV) while ensuring generalization across visual scenes, in simulation as well as in the real world.

**Evaluation protocol.** A single test run consists of initializing a scene with a number of objects in the drone's FOV and providing text instructions about the goal to reach. The closed-loop inference is run for a fixed number of steps, 80, slightly larger than the average training sequence length. The test is successful if the agent can navigate towards the user-instructed object and center it in the middle of the frame. Failures, on the other hand, can be identified when the drone loses the object (the target exits the FOV and/or another visual cue is centered in on), or fails to approach or center in on the goal. The evaluation of the closed-loop performance of our system is based on monitoring the success rates on repeated runs in various evaluation configurations.

### 4.2 EXPERIMENTAL SETUP

#### 4.2.1 SIMULATION

**Simulator.** We use the PyBullet physics engine, building off the codebase presented in (Panerati et al., 2021). The drone dynamics we use are based on Bitcraze's Crazyflie 2.x nano-quadrotor. The physics simulation runs at 240Hz, and we allow low-level flight control to occur at the same frequency, although inference runs at a much slower rate of 3Hz. Indeed, for inference or data collection, we simulate the evolution of the system with constant commands for the number of steps corresponding to the desired period.

**Background scenes.** In addition to the in-distribution (InD) scene which we use for training, Samurai, we design a second very different-looking environment; Stadium. In stark visual contrast to the training environment which has tiled flooring, the Stadium environment ground is covered by green textured grass and field lines. Also, the Stadium stands include large portions of purple walls and its structure is distinctly different from that of the Samurai temples. The Stadium environment is used for scene generalization evaluation in simulation (see Figure 11).

**Test scenarios.** Each feature extractor, policy head, and dataset combination considered is tested in simulation on an increasingly demanding suite of scenarios. In each case, we gather the success rates over multiple runs, randomly initializing the positions of the potential target goals, the quadrotor distance to the objects, and the initial heading angle. Each scenario is run in both the InD (Samurai)

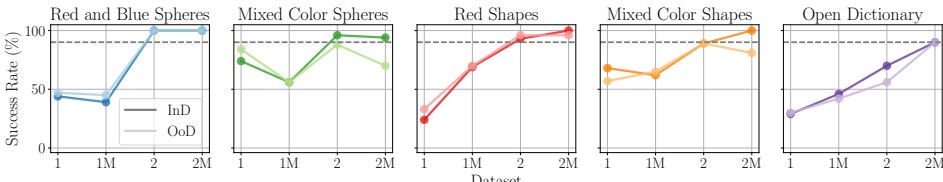

Figure 2: Success rate as a function of the dataset richness on all five simulation test scenarios. Darker lines correspond to the InD scene, and lighter colors to the OoD background. Each data point is obtained from 100 runs with command syntax "Navigate to the [OBJECT]".

and OoD (Stadium) scenes. The breakdown of scenarios considered (depicted in Figure 11) is as follows:

*1. Red and Blue Spheres:* Easiest setup providing a measure of the mastery of the unaltered training task and performance changes based only on the change of scene and/or instruction phrasing.

*2. Mixed Color Spheres:* Tests the generalization capability with respect to colors with a choice of two out of red, blue, green, yellow, and purple spheres appearing at initialization.

*3. Red shapes:* Evaluates the sensitivity and adaptability to shapes of same color (red) with two out of a sphere, a cube, and a pyramid positioned in the quadrotor's initial FOV.

*4. Mixed Color Shapes:* Similar to above with the object colors also randomized to be any of red, blue, green, yellow or purple.

*5. Open Dictionary:* Hardest setup that goes beyond shapes and colors, with a range of objects in a more cluttered scene. Three objects are placed in the drone FOV picked amongst a red sphere, blue sphere, a light-colored Jeep, an Australian cattle dog , a brown Horse, a tall and narrow Palm Tree, a toy Space Rocket, and a whole Watermelon.

### 4.2.2 REAL-WORLD TRANSFER

**Hardware.** Our setup utilizes a DJI M300 RTK quadcopter interfaced with a DJI Manifold 2 computer and the DJI Onboard SDK, processing commands on a base station via Wifi to achieve a runtime frequency of just over 1 Hz with our highest resolution models. Flight tests are conducted on an urban university campus lawn, with targets including various cardboard cutouts positioned on tripods. More details are provided in Appendix A.4.

**Test setup.** We deploy the system with the ViT policy head on the drone hardware in a series of tests with various props as targets and in different two-object initial configurations, in an urban campus environment. This is the ultimate challenge exposing the agents simultaneously to sim-to-real transfer, new scene generalization (see Figure 9), as well as new object instruction handling.

### 4.3 SUMMARY OF GENERALIZATION TESTS

We tested the model's generalization capabilities across both environments and objects. The model was trained in a single simulated environment (Samurai) using only two spherical objects (blue and red) and evaluated in three scenarios: the same training environment, a new simulated environment (Stadium), and a campus lawn in the real world. Despite the very limited training data, the model generalized well to open-set scenarios, including objects with varying shapes and colors, a wide range of simulated objects (e.g., a Jeep, a horse, a palm tree, and a watermelon), and real-world objects (e.g., a yellow star, a cutout of a man with a wig) during drone navigation.

## 5 RESULTS

### 5.1 DATASET DESIGN

The degree of dataset richness required for generalisation is evaluated by training `Flex` instances of 256-patch resolution and a ViT policy head on all of the four datasets described in Section

3.1. The success rates on the simulation test cases are depicted in Figure 2. There is a clear gap in performance between models trained on single object examples and two objects. Indeed, the former systematically try to reach the red ball when present regardless of the text instruction. They also generalize significantly worse to open dictionary objects (under 50% success rate). The gains from instruction augmentation are less potent, especially on simple geometries and color variations. However, the open dictionary setup suggests that augmentation, which acts both on the action but also choice of noun for the goal object, can offer interesting performance benefits. Thus, we conclude that training with two goal objects along with instruction augmentation (which is cheap to implement as described in Appendix A.3) are recommended practices to attain satisfactory generalization.

## 5.2 FEATURE EXTRACTION RESOLUTION

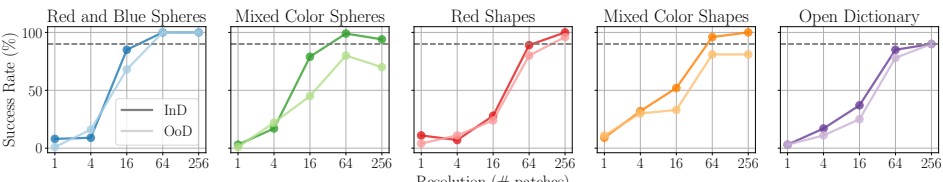

Figure 3: Success rate as a function of the feature extractor resolution on all five simulation test scenarios. Darker lines correspond to the InD scene, and lighter colors to the OoD background. Each data point is obtained from 100 runs with command syntax "Navigate to the [OBJECT]".

A central claim in this work is that simply using a pre-trained VLM as a text and whole image encoder is not suitable for robotics applications. This hypothesis, along with the question of the resolution of spatial features required to achieve robust visual navigation is investigated by training five **Flex** instances with a ViT policy head on the 2M dataset. We increase patch resolution from a single patch containing the entire image to the BLIP-2 limit of 16×16 non-overlapping square patches. The performance in each case on our test suite is provided in Figure 3. The results corroborate the claim regarding the failure of entire image processing. Indeed, there is a definite pattern of performance improvement with patch resolution, with a minimum of 64 patches (8×8 grid division of the input frame) needed to guarantee generalization close to the 90% mark on all test setups.

## 5.3 POLICY NETWORKS

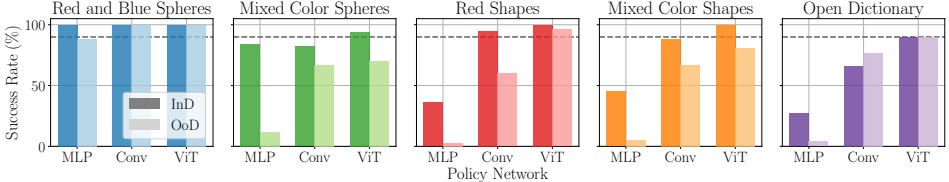

Figure 4: Success rate as a function of the policy head architecture on all five simulation test scenarios. Darker lines correspond to the InD scene, and lighter colors to the OoD background. Each data point is obtained from 100 runs with command syntax "Navigate to the [OBJECT]".

**Performance.** Policy heads, the only trainable component of **Flex**, are a crucial factor for the quantitative performance of the agents, but also dictate the of trajectories and behavior exhibited by the closed-loop navigation system. The former is tested on 256-patch full resolution models trained on the 2M dataset for all four policy architectures considered (presented in Section 3.3). Results are provided in Figure 4. Basic MLP policies, though capable of achieving the original training task both in and out-of-distribution, suffer from a drastic loss in performance on all but the mixed color spheres task InD, with close to complete failure in the OoD setting. This indicates that the VLM patch-wise features are not sufficiently simple and universal for direct mapping into correct decision commands. Thus, an important role in useful task information retrieval has to be played by an adequate policy architecture. Both the Conv and ViT policies offer somewhat strong performance. The latter consistently surpasses the former by 10 % or more, and with dips below the

90% performance bar only in OoD settings where confusion between purple goals and the purple background occur.

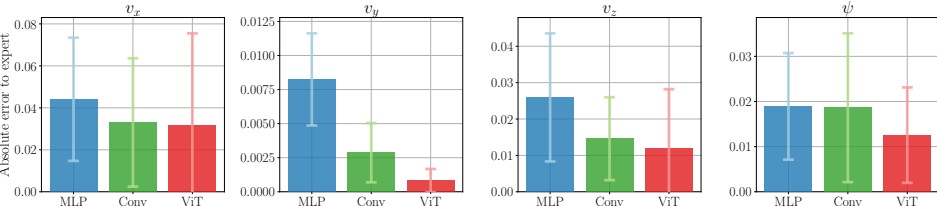

Figure 5: Absolute error to expert per policy network for each of the output dimensions ($v_x$, $v_y$ and $v_z$ are in m/s while $\dot{\psi}$ is in rad/s). Each data point is obtained from 22.5k frame-instruction pairs.

**Flight behavior.** Stark differences in closed-loop behavior are observed: the MLP policy leads to very erratic closed-loop navigation and shows reluctance to stop at the goal, the Conv head exhibits aggressive piloting resorting to abrupt turns in front of goals, while ViT offers much smoother trajectories closer to those seen in training. To back these observations, we generate a total of 30 expert demonstration runs (sphere, cube and pyramid goals with all five color variations in both InD and OoD scenes), that are identical in the initial placement of the target and expert 150-command sequence. We run the models on the expert frames with five text variations of the text instruction including commands in French and Italian (see Appendix A.6). The difference between each scalar output and its corresponding expert command for all objects, scenes, instructions and sequence frames is depicted in Figure 5. The figures seem to corroborate the qualitative observations, showing significantly smaller deviations from expert decisions with the ViT policy ($\sim$40% better than its counterpart on the crucial yaw rate command $\dot{\psi}$), and smoother flight control ($3\times$ and $8\times$ smoother than Conv and MLP on sideways crabbing $v_y$). Thus, we empirically establish ViT's superiority as a policy head in terms of generalization performance as well as flight behavior.

## 5.4 ROBUST ViT DECISIONS

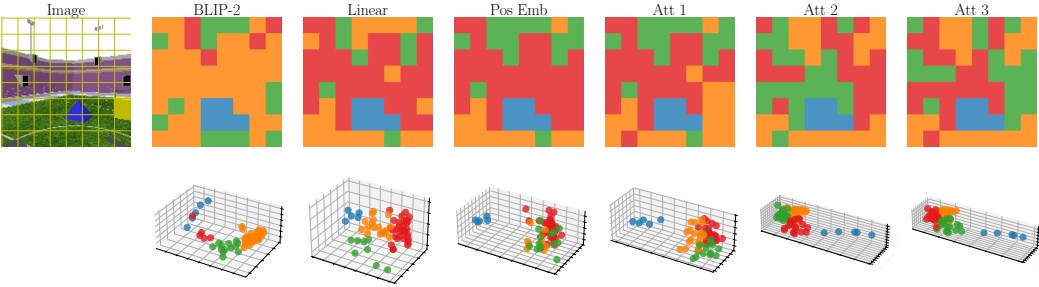

Figure 6: Feature clustering and visualization through the 64-patch ViT policy network. The instruction is "Navigate to the blue pyramid" with a frame (top left with a grid overlay separating the patches) from the OoD simulation scene. The top row depicts the cluster memberships by color, with the goal belonging to blue. The bottom row visualises the features' t-SNE embeddings in 3D.

**Similarity-based clustering and visualization.** The ViT policy offers a structured representation of features across the network as per patch feature spatial attribution is respected up until the last linear decision layer. We leverage this structure and apply similarity-based clustering of features to elucidate the decision process. Indeed, at a given layer, we first L2-normalize all patch-wise features before applying $k$-means clustering ($k$ selected via the "elbow" technique). We note that $k$-means minimizes intra-cluster variances, hence acts on squared Euclidean distances. For visualization, we apply the t-distributed stochastic neighbor embedding method (t-SNE) in 3D to the normalized features, using the squared Euclidean distances as our metric (cf. Algorithm 1 in Appendix A.7). The choice of metric is motivated by the proportionality of the square Euclidean distance to the cosine distance for L2-normalized vectors. Thus, we ensure consistency in both clustering in the original feature space

based on cosine distance and the preservation of local similarity structure for visualization. Using Figure 6 for illustration, the pyramid cluster accurately espouses the approximate region of the goal for all layers. However, the t-SNE projections seem to show that goal (blue) and background (other) features are not clearly separable from the start (BLIP-2 extraction level). Visually, we conjecture that, as the features are transformed by the attention layers, the non-essential background features become both increasingly indistinguishable from each other and dissimilar to the goal patches, with growing margin for a clear decision boundary to leverage only task related information.

**Cluster separability scoring.** We tailor the global clustering Davies-Bouldin Index (DBI) (Davies & Bouldin, 1979), to associate it only with the cluster in which the frame patches intersecting the goal object lie, or goal cluster for short (if applicable, the cluster is picked by majority number of members). Whereas the original index is the average similarity measure of each cluster with its most similar cluster, we take only the highest pairwise similarity score between the goal cluster and the others. Here, similarity is defined as the ratio of intra-cluster distances to inter-cluster distances (details in Appendix A.7). Thus, we obtain a metric that

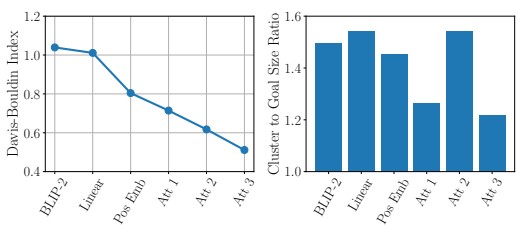

Figure 7: DBI and cluster to goal patch size ratio geometric averages across the 64-patch ViT network layers. Each data point is averaged from 30 runs of 150 frames ($N = 4.5$k).

favours configurations in which the cluster of interest is less dispersed and farther apart from others, with lower values indicating higher separability of the cluster. Figure 7 depicts the geometric means of the DBI and the ratio of goal cluster to target size across network layers on frames from 30 runs with various goal objects and scenes. It clearly supports the claim that through the ViT layers, the goal cluster contains mostly target patches and is increasingly separated from the rest of the features for subsequent linear mapping to commands. This robust decision mechanism appears to be invariant for various scenes, goal objects, and instruction formulations.

## 5.5 REAL-WORLD DEPLOYMENT

`Flex` (2M Dataset, 256-patches, ViT policy) transfers seamlessly to the real world and gracefully handles a variety of new scenarios. Indeed, the system exhibits highly robust performance of the task on the outdoor campus lawn, with new unseen objects as goals and various backgrounds and lighting conditions, with no notable failures. Frames from an example run can be seen in Fig 8 (successful runs for six other different goals are depicted in Fig 9 of Appendix A.6).

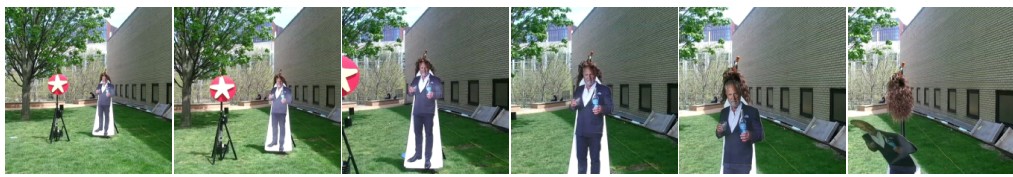

Figure 8: `Flex` sample real test run: Frames from a test run with text instruction "Fly to the man with a wig". Time increases from left to right. In the last frame, the cardboard cutout is blown off the tripod support by the drone propellers. The wig remains.

## 6 RELATED WORK

**End-to-end robot learning.** End-to-end deep learning has shown significant potential in autonomous navigation tasks (Chib & Singh, 2023; Bojarski et al., 2016; Pomerleau, 1988). Advances in safety (Xiao et al., 2023) and generalization (Chahine et al., 2023; Quach et al., 2024; Wang et al., 2023b; Yin et al., 2023; Kaufmann et al., 2023) have improved performance, but these models remain largely black-box, incapable of user interaction, and confined to the scope of training data. Moreover, training robust, large-scale models is challenging due to the need for extensive, high-quality datasets, which are costly, time-consuming, and pose potential safety risks (Kendall et al., 2019).

Simulation-based training has emerged as a practical alternative, leveraging platforms such as VISTA (Amini et al., 2022), Drake (Tedrake et al., 2019), PyBullet (Panerati et al., 2021), and AirSim (Shah et al., 2018). However, simulated environments often fail to fully capture real-world intricacies, leading to performance degradation and safety risks during deployment. Intermediate visual abstractions (Müller et al., 2018; Toromanoff et al., 2020; Behl et al., 2020) address some of these gaps, but such methods lack the multimodal reasoning required for truly generalizable systems.

**VLMs and Foundation Models in Robotics.** Foundation models, particularly vision-language models (VLMs), have revolutionized open-world visual understanding tasks, including classification (Radford et al., 2021; Yang et al., 2022), detection (Li et al., 2022c; Zhong et al., 2022), segmentation (Kirillov et al., 2023; Li et al., 2022a), and captioning (Li et al., 2023; Wang et al., 2022). Within robotics, these models have been applied to open-vocabulary detection and manipulation (Chen et al., 2022; Liu et al., 2024), planning (Ahn et al., 2022), and action prediction (Brohan et al., 2023). For navigation, approaches that decouple perception and control (Maalouf et al., 2023) or generate waypoints explicit (Shah et al., 2023) have been proposed.

In dynamic, open-set environments, VLMs have facilitated applications like 3D mapping (Huang et al., 2023; Ding et al., 2023), scene segmentation (Peng et al., 2023; Jatavallabhula et al., 2023), and explainable, language-based representations (Kim et al., 2019; Omeiza et al., 2021; Kuo et al., 2022; Tan et al., 2023; Zhong et al., 2023). However, despite their versatility across data modalities (Ramesh et al., 2021; Crowson et al., 2022; Patashnik et al., 2021; Ramesh et al., 2022), these methods often rely on modular pipelines and global embeddings, which limit their utility for text-instructed end-to-end robotic learning.

**`Flex` vs. Mainstream VLN Approaches.** Recent advances such as RT-1 (Brohan et al., 2022), RT-2 (Brohan et al., 2023), and Vint (Shah et al., 2023) represent significant progress in vision-based navigation. RT-1 was trained on over 130,000 real-world demonstrations, while RT-2 incorporated internet-scale pre-training with models up to 55 billion parameters. Similarly, VLN-BERT (Hong et al., 2021) was trained on more than six million image-text-action triplets, and NavGPT (Zhou et al., 2024) leverages GPT models for zero-shot action prediction. In stark contrast to our minimalist approach training small policy heads on relatively tiny amounts of data, these methods rely on extensive datasets and resource-intensive training pipelines.

**Text-Patch Features for End-to-End Robotics.** Patch-based feature extraction has been explored in prior work, but existing methods face limitations. Some are not multimodal (Amir et al., 2021); others fine-tune encoders for 2D-pixel alignment, losing critical concepts (Ding et al., 2022). Approaches like SAM (Kirillov et al., 2023) rely on segmentation models that can miss important regions (Jatavallabhula et al., 2023; Maalouf et al., 2024), while others fail to fuse text queries with patch descriptors for semantic relations (Wang et al., 2023a).

Our approach bridges these gaps by extracting fine-grained, text-fused features from pre-trained VLMs, enabling context-aware reasoning critical for end-to-end robotics tasks without relying on hand-designed pipelines or intermediate representations (Li et al., 2022b; 2023; Radford et al., 2021).

## 7    CONCLUSION

This work establishes the essential dataset and model requirements for robust generalization in text-instructed end-to-end visual navigation agents using pre-trained VLM encoders as multi-modal feature extractors. Our findings include the failure of training on a single data context (leads to over-fitting), and the adequacy of two examples to train models that handle a wide spectrum of similar use cases. We also advocate for simple text-space augmentations, which can improve performance in more nuanced test settings. We shed light on the shortcomings of low-resolution patch-wise feature extraction, with the fly-to-target task necessitating at least $8\times8$ patches. Finally, we ascertain the superiority of the ViT architecture as a policy head, in terms of task success and flight behavior, while uncovering aspects of its robust context invariant decision process via similarity-based clustering.

The synthesis of these findings is **`Flex`**, a new minimalist training framework capable of producing user-interactive highly generalizing visual navigation agents. Our solution elegantly handles a suite of in-simulation challenges and proves readily deployable in the real-world, robustly achieving direct sim-to-real open dictionary out-of-distribution generalization.

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

# A APPENDIX

## A.1 EXAMPLE FOR EXTRACTING $m \times m$ RESOLUTION DESCRIPTORS

In this example, assume $w' = 4$ (i.e., we have 16 patches in total), and the desired resolution is $2 \times 2$ ($m = 2$). The original grid of patches is denoted by:

$$\begin{bmatrix} p_{1,1} & p_{1,2} & p_{1,3} & p_{1,4} \\ p_{2,1} & p_{2,2} & p_{2,3} & p_{2,4} \\ p_{3,1} & p_{3,2} & p_{3,3} & p_{3,4} \\ p_{4,1} & p_{4,2} & p_{4,3} & p_{4,4} \end{bmatrix}.$$

The Coarser grid (of sub-grids) is given by:

$$\begin{bmatrix} \begin{bmatrix} p_{1,1} & p_{1,2} \\ p_{2,1} & p_{2,2} \end{bmatrix} & \begin{bmatrix} p_{1,3} & p_{1,4} \\ p_{2,3} & p_{2,4} \end{bmatrix} \\ \begin{bmatrix} p_{3,1} & p_{3,2} \\ p_{4,1} & p_{4,2} \end{bmatrix} & \begin{bmatrix} p_{3,3} & p_{3,4} \\ p_{4,3} & p_{4,4} \end{bmatrix} \end{bmatrix}.$$

Finally, to extract a descriptor for each of these 4 subgrids, we use 4 calls to our methods with 4 different masks, each mask corresponding to a subgrid. The masks are given by the row stacking of these matrices:

$$\begin{bmatrix} 1 & 1 & 0 & 0 \\ 1 & 1 & 0 & 0 \\ 0 & 0 & 0 & 0 \\ 0 & 0 & 0 & 0 \end{bmatrix}, \begin{bmatrix} 0 & 0 & 1 & 1 \\ 0 & 0 & 1 & 1 \\ 0 & 0 & 0 & 0 \\ 0 & 0 & 0 & 0 \end{bmatrix}, \begin{bmatrix} 0 & 0 & 0 & 0 \\ 0 & 0 & 0 & 0 \\ 1 & 1 & 0 & 0 \\ 1 & 1 & 0 & 0 \end{bmatrix}, \begin{bmatrix} 0 & 0 & 0 & 0 \\ 0 & 0 & 0 & 0 \\ 0 & 0 & 1 & 1 \\ 0 & 0 & 1 & 1 \end{bmatrix}.$$

## A.2 DISCUSSION

**Limitations and Future Work** The framework presented in this manuscript is limited to instantaneous decisions. Indeed, the policy can only act with information from the current image and has no access to a history of representations or actions. We are keen to incorporate our potent multi-modal feature encoding scheme into sequential decision-making processes. This would enable **Flex** to go beyond generalization between environments and objects, and handle instructions over actions, sequences of steps, and behavior modes. An additional limitation of this work is its computational overhead, which renders it impractical for real-time execution on small mobile robotic platforms. This pertains to the wider effort in edge AI research to enable the deployment of foundation models directly on edge devices. The robotics community will reap great benefits from these advances that will enable the widespread adoption of methods such as **Flex**.

## A.3 TRAINING

**Dataset description.** A unique simulated dataset is used for training by all models discussed throughout this paper. It consists of 300 goal approach flight sequences (22,844 frames in total, 76.15 frames per run on average), with a 90% training and 10% validation split. The same two objects, a red and a blue sphere, are used in every sequence. All frames of a give sequence are labelled with the same text, a natural language instruction sentence providing information on whether to fly to the red or blue goal. We balance the number of runs headed for each of the two possible options.

**Trajectory design.** Both spheres are initially positioned in the drone's field of view such that all trajectories generated carry recovery information (with the farthermost target at the border of the image). They are positioned at the same altitude and relative position, equidistant from the drone. We randomize the initial distance to the targets, thus ensuring the size of the objects in the image is varying. This ensures we expose the network to trajectories that recenter and approach the goal target from a wide spectrum of angles and distances. The control signals are obtained with the ground truth knowledge available to us from the simulator, where PID controllers generate the vertical velocity commands $v^z$ to reduce the altitude gap, the forward velocity commands $v^x$ to reduce the distance gap, and the yaw rate $\dot{\psi}$ to pilot the drone towards the instructed goal (centered in the middle of the frame).

**Label design.** Each approach sequence is associated with a unique text instruction. We introduce text domain augmentation by generating 25 synonyms to the verb "fly to" and noun "target". The verb phrases include: migrate towards, glide to, whiz to, steer towards, manoeuvre to, zoom to, elevate towards, approach, propel to, make way to, orient towards, venture towards, soar to, advance to, progress towards, journey to, hover to, proceed to, drift towards, rush to, shift to, head towards, ascend towards, scene towards, travel to. The object terms include: signpost, point, terminus, stage, station, location, interest, goal, setting, checkpoint, cue, objective, coordinate, emblem, locus, target, marker, beacon, spot, destination, signal, symbol, sight, position, aim. We uniformly sample from these terms to form instructions in the format: *VERB PHRASE* the [COLOR] *OBJECT*

Training run sequence frames are depicted in Figure 10.

**Training details.** The loss used is the Mean Squared Error (MSE) between the network predicted commands and the values with which a dataset frame is labeled, with equal weights between all scalar outputs. We train the models for 20 epochs, using the Adam optimizer with a learning rate of $1 \cdot 10^{-4}$. Frames are uniformly sampled from the dataset to ensure shuffling. We take the checkpoint with the best validation loss for each model. All training was performed on a single NVIDIA GeForce RTX 3080 Ti GPU, with 12 GB memory and 10240 CUDA cores, with a single full training run taking around 45 hours. The major compute bottleneck originates in repeated calls to the BLIP2 based multi-modal patch encoding, which can be alleviated by encoding and storing the entire dataset features once instead of re-encoding at every epoch. Our setup is capable of handling 2.8 frames per second on average during training.

### A.4 Real world setup details

**Hardware.** Our platform is a DJI M300 RTK quadcopter. The M300 interfaces with the DJI Manifold 2 companion computer, enabling programmatic control of the drone. The DJI Onboard SDK (Software Development Kit) and its associated ROS wrapper provide an interface for feeding the drone's low-level flight controller with desired high-level translation velocities and yaw rate commands. The flight controller onboard the DJI M300 is a black box system provided by the manufacturer which controls the four-rotor speeds to track the velocities specified by the companion TX2 computer. It is worth mentioning that the dynamics of the platform we use do not match those of the nano-quadrotors simulated. Input images gathered by the gimbal-stabilized camera, which follows the drone's yaw to always point forward, are available to the companion computer via the SDK. The onboard computer runs an NVIDIA Jetson TX2, which has GPU capability. However, `Flex` inference on a single image takes over 10 seconds. Thus, establish a connection over Wifi between the onboard TX2 computer and a standalone Lenovo 16" ThinkPad P16 Gen 2 with Intel Core i9-13980HX (13th Gen) CPU and NVIDIA RTX 5000 GPU with 16 GB GDDR6 VRAM. The TX2 sends the latest image to the machine which runs inference and replies with the control command to execute. We reach a runtime frequency of just over 1 Hz with our setup.

**Real world scene.** We conduct our flight tests on an outdoor lawn in an urban university campus. In addition to having to bridge the sim-to-real transfer gap, agents are also exposed to a completely new visual scene, with various buildings, reflective structures, and trees. Lighting conditions from various starting positions now expose the agents to sunlight from various angles, making for a very challenging sim-to-real generalization scenario.

**Goal objects.** We use cardboard cutouts that we position on tripods at a safe flight altitude as targets. The list of props contains a red disk, blue disk, white disk, yellow square, yellow star, and human figure printed cutout on top of which a wig is placed.

### A.5 Model Details and Parameters

**Policy models** We train four different policy head architectures: A **Vision Transformer (ViT)** architecture consists of three Transformer blocks with a patch size of 1x1, a dimensionality of 128, and four attention heads. The multilayer perceptrons (MLPs) in this model have a dimensionality of 256. A fully-connected layer maps to a 4-dimensional output. **The convolutional network (Conv)** architecture includes three 1x1 convolutional layers. The first layer has an input dimension of 64, and all layers have a hidden dimension of 128. Each layer is followed by ReLU activation and dropout, and the output is flattened to (128 * 16 * 16) and a fully-connected layer maps to a 4-dimensional

output. Finally, the **Multilayer Perceptron (MLP)** architecture begins by average pooling the [B, 64, 16, 16] patches into [B, 64, 1, 20]. It then applies a fully-connected layer with dimension 1280, using ReLU activation and a dropout rate of 0.3. A fully-connected layer maps to a 4-dimensional output.

Table 1: Vision Transformer Policy Parameters

| Parameter | Value |
|---|---|
| Image Size | 32x1 |
| Patch Size | 1x1 |
| Number of Classes | 4 |
| Dimension | 128 |
| Depth | 3 |
| Heads | 4 |
| MLP Dimension | 256 |
| Channels | 64 |
| Dimension per Head | 32 |

Table 2: Convolutional Network (Conv) Policy Parameters

| Parameter | Value |
|---|---|
| Number of Layers | 3 |
| Hidden Dimension | 128 |
| Activation Function | ReLU |
| Dropout | 0.3 |

Table 3: Multilayer Perceptron (MLP) Policy Parameters

| Parameter | Value |
|---|---|
| Pooling | Average Pooling |
| Pooled Dimension | [B, 64, 1, 20] |
| FC Layer 1 Dimension | 1280 |
| Activation Function | ReLU |
| Dropout Rate | 0.3 |
| FC Layer 2 (Output) Dimension | 4 |

## A.6    ROBUSTNESS TO SYNTAX FORMULATIONS

**Testing alternative commands .**  We generate a set of five text command variations for each object amongst the sphere, cube and pyramid and colors blue, red, green, purple and yellow, in order to test the robustness of models to syntax and language formulations. The skeleton of each command (left) and an example for the green sphere (right) are given below:

- Fly to the [OBJECT]
- Navigate to the [OBJECT]
- Reach the [OBJECT]
- Vole vers [OBJECT IN FRENCH]
- Vola verso [OBJECT IN ITALIAN]

- Fly to the green ball
- Navigate to the green ball
- Reach the green ball
- Vole vers la balle verte
- Vola verso la palla verde

## A.7    FEATURE CLUSTERING AND ANALYSIS

We provide the algorithms used to perform feature clustering and visualization (Algorithm 1) and compute the cluster score (Algorithm 2).

---

**Algorithm 1:** Feature Clustering Analysis Algorithm

---

**Model :** End-to-end network $\phi$
**Input:** Text instruction $T$, Frame $F$, Goal patch indices $I$
**Result:** Goal cluster score for each layer

```
1  X ← {F, T}                              // Initialize state with input
2  n ← Num-Layers(φ)                       // Get the number of network layers
3  n ← n − 1                               // Ignore ultimate decision layer
4  DBI ← zeros(n)                          // Initialize layer scores
5  for i ∈ [0, ..., n − 1] do

      // Clustering
6     X ← φᵢ(X)                            // Forward pass through iᵗʰ layer
7     X̄ ← L2-Normalize(X)                  // L2 Normalize features
8     k ← Elbow(X̄)                         // Optimize number of clusters
9     C ← k-means(X̄, k)                    // Obtain k-means clustering
10    c ← Argmax-Members(C, I)             // Identify cluster with most goal
       patches
11    DBI[i] ← Compute-DBI(X̄, C, c)        // Compute DBI for goal cluster

      // Dimensionality reduction
12    D ← Squared-Pairwise-Dists(X̄)        // Compute the squared
       distances
13    X̄₃D ← t-SNE(D)                       // Reduce to 3D via t-SNE
14    Vis(X̄₃D, C)                          // Visualize with original clustering
15 return DBI
```

---

**Algorithm 2:** Compute-DBI

---

**Input:** Normalized features $\bar{X}$, $k$-means clustering $C$, Cluster index $c$
**Output:** Separability score for cluster $c$

```
1  G ← Get-Centroids(X̄, C) ;                    // Get the cluster centroids
2  D_inter ← Pairwise-Dists(G) ;                 // Compute inter-cluster distances
3  d_max ← Max(D_inter) ;                        // Maximum inter-cluster distance
4  D_intra ← Intra-Clust-Dists(X̄, C) ;          // Compute intra-cluster distances
5  D_inter, D_intra ← D_inter/d_max, D_intra/d_max ;   // Normalize distances
6  n ← Get-Num-Clust(C) ;                        // Get the number of clusters
7  c-sim ← zeros(n) ;                            // Initialize similarity scores
8  for i ∈ [0, ..., n] \ {c} do
9      c-sim[i] ← (D_intra[c] + D_intra[i])/D_inter[c, i] ;   // Compute i to c similarity
10 return Max(c-sim) ;                           // Return maximum score
```

Table 4: Model Parameters Breakdown

|  | ViT | Conv | MLP |
|---|---|---|---|
| Total Model | 1,173,055k | 1,172,822k | 1,172,654k |
| VLM Encoder | 1,172,600k | 1,172,600k | 1,172,600k |
| Last Layer Projection | 49k | 49k | 49k |
| Trainable Params | 454k | 222k | 54k |
| Policy network | **405k** | **172k** | **5k** |

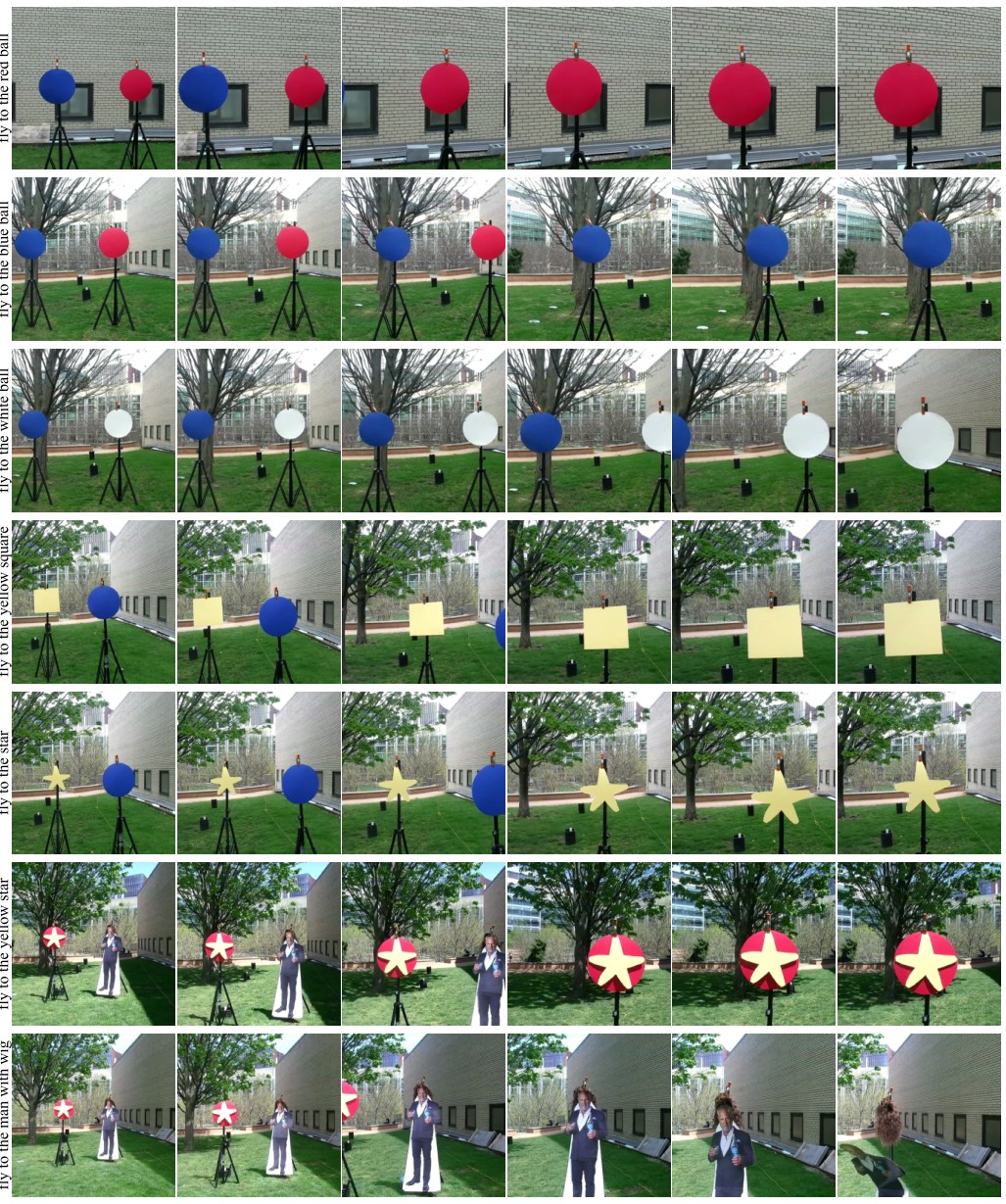

Figure 9: `Flex` in the wild: screenshots of test runs with the ViT policy network (one per row, time increases from left to right) performed on a lawn on the urban campus with various goal objects, backgrounds and lighting conditions. The text instruction used in each case is by the first image.

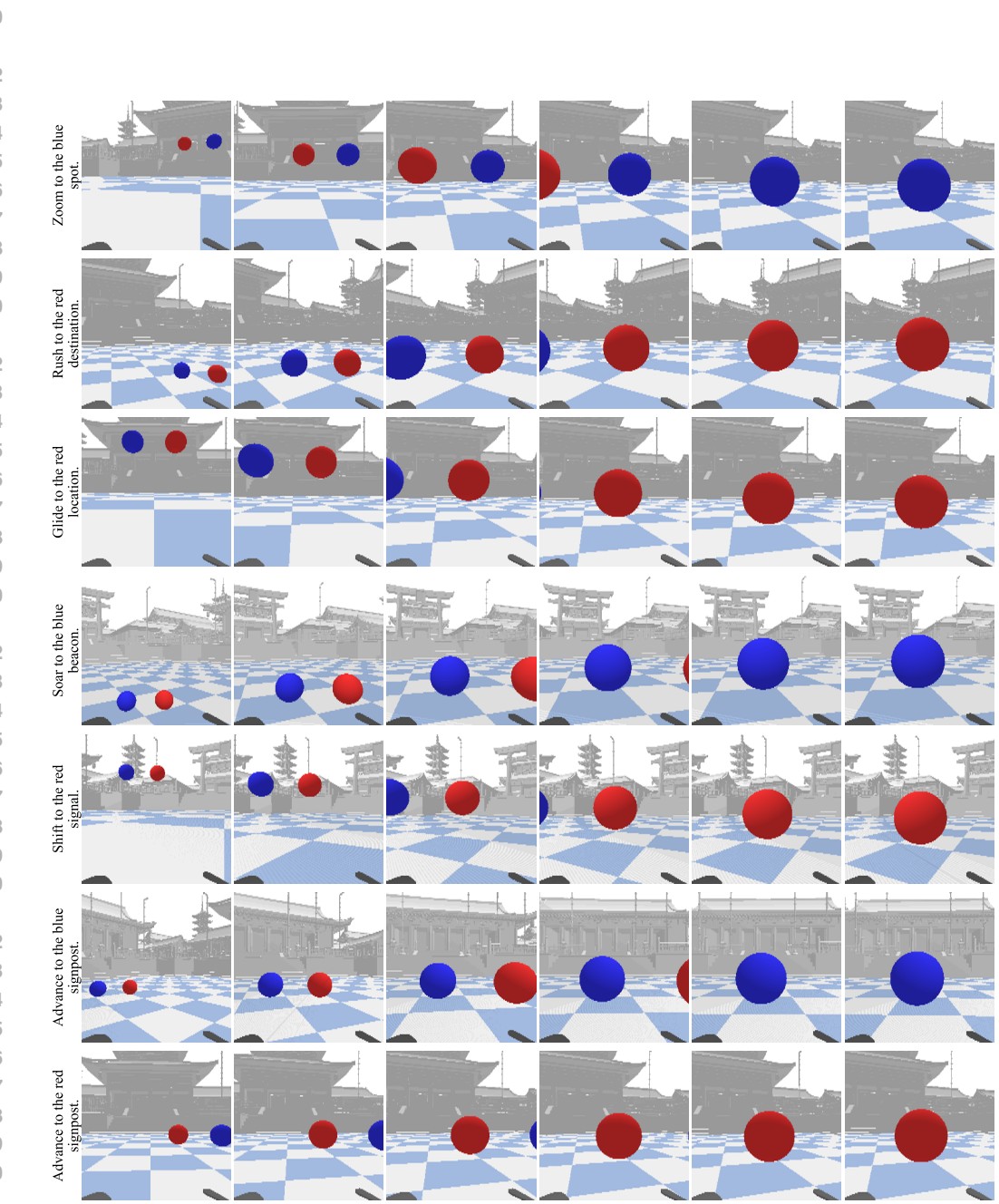

Figure 10: 2M training examples

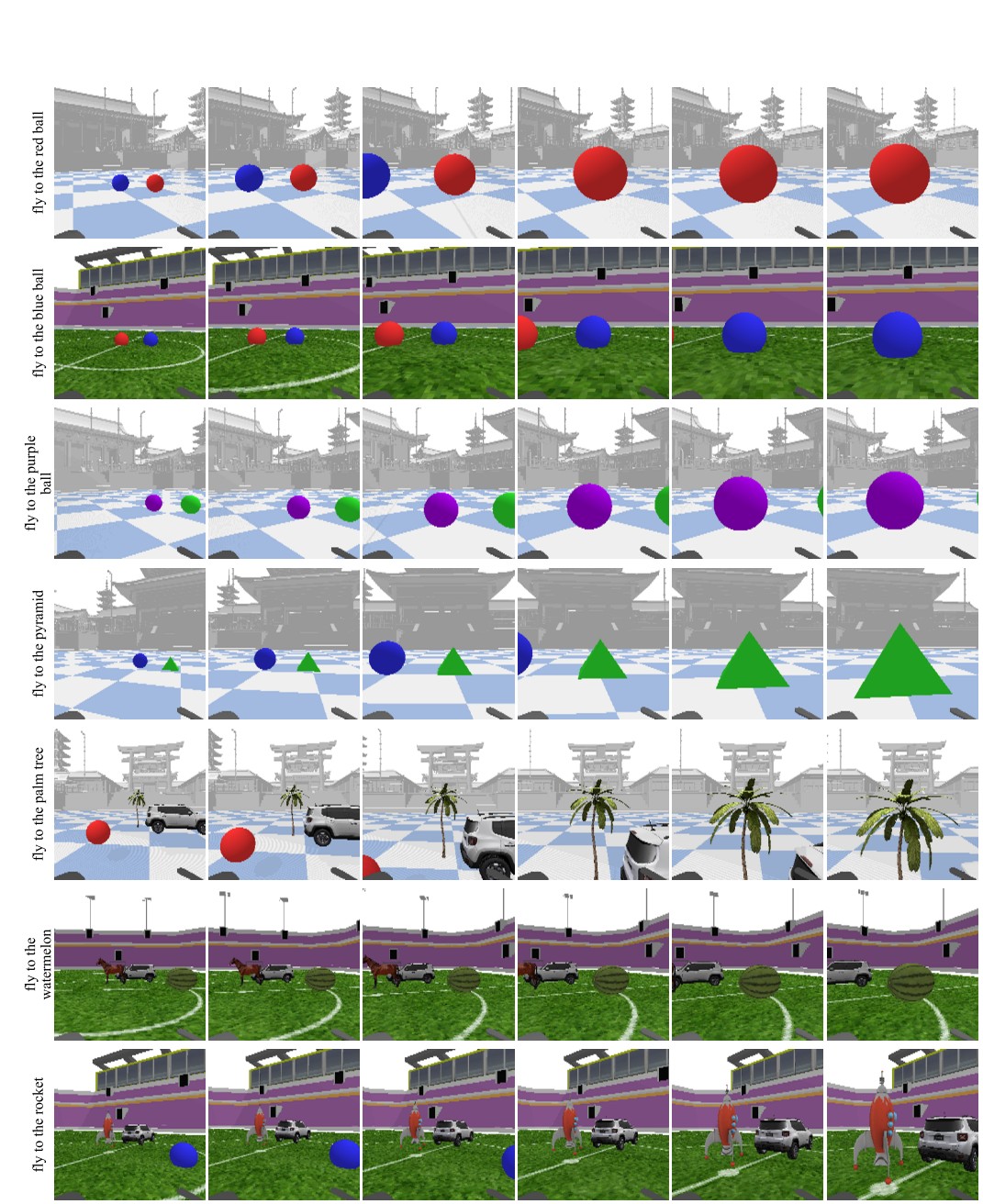

Figure 11: 256-patch ViT closed-loop inference in simulation

