# OpenReview forum: "Flex: End-to-End Text-Instructed Visual Navigation with Foundation Models"
_ICLR.cc/2025/Conference — Submitted to ICLR 2025_

### Official Review · Reviewer_BS4x · 2024-11-04

**Soundness:** 2
**Presentation:** 1
**Contribution:** 2
**Rating:** 3
**Confidence:** 4

**Summary:**

This paper conducts an empirical study on using features extracted by frozen Vision-Language Pretrained Models (VLMs) for downstream policy learning in object-guided quadrotor visual navigation. The policy is trained with synthetic data in simulation and demonstrates a real-world deployment example, providing an initial test of generalizability.

**Strengths:**

1. The motivation to investigate frozen VLM encoders for semantically rich and spatially aware embeddings without extensive retraining is clear and well-grounded, offering a promising approach to simplify training in complex robotic tasks.
2. The experimental setup is thorough, exploring various dataset configurations and policy architectures, which provides valuable insights into design choices for multi-modal robotic policies.

**Weaknesses:**

1. Limited Task Scope: While the use of VLM features for visual navigation is compelling, the experiments are restricted to quadrotor object navigation. Broader experiments across different robotic tasks or navigation scenarios could help demonstrate Flex’s general applicability, aligning with the paper’s claims of “closed-loop visual navigation agents that generalize across unseen environments.”
2. Feature Extractor Generalizability: Although the study aims to investigate “suitable feature extractors for text and vision-based robotics learning,” the experiments focus solely on BLIP-2 and modifications to its Q-former. Including other VLMs, such as CLIP or ImageBind, would strengthen the claim of generalizability and demonstrate the adaptability of the proposed method across diverse VLM architectures.
3. Task Complexity: In the current setup, the target is observable from the beginning, and language instructions are relatively simple. Visual navigation in complex settings involves learning semantic associations between language and visual cues and exploring the environment effectively to locate and reach the target. Here, the task primarily involves target identification and navigation, with limited indication of the benefits of end-to-end learning over modular approaches like GOAT. Clarifying the advantages of end-to-end learning in this setup would improve the paper’s impact.
4. Language Instruction Scope: The difficulty and complexity of language instructions in the dataset are not discussed. Unlike works in Zero-Shot Object Navigation (ZSON) or Zero-Shot Vision-and-Language Navigation, which leverage the language comprehension capabilities of VLMs, the findings in this paper are limited to basic instructions. This limitation makes it challenging to support claims of Flex being “user-interactive,” as the current level of interaction falls short of handling more complex language instructions effectively.

**Questions:**

See weakness

---

> ### Author Response · Authors · 2024-11-19
> **Initial reply**
>
> We sincerely thank the reviewer for their dedicated and meticulous evaluation of our paper. Their thoughtful insights and careful reading have been pivotal in refining our work. We have diligently addressed each of their valuable concerns and are eager to engage in further discussions to resolve any remaining issues.
>
> ## Weakness 1
>
> The reviewer makes a good point in inscribing the work in the larger scheme of robot visual navigation. We argue that the admittedly simple task does not limit the significance of the findings and does not contradict the quoted assertion.
>
> ### Platform-Agnostic Pipeline
> First, the entire pipeline is completely platform agnostic, except for specifying the action space. The 4-dimensional action space used here is, in fact, more complex than the case of driving (often 2D with steering angle and acceleration). We know from related work that the pipeline presented will translate seamlessly (see [1] as an example). This also applies to humanoids and quadrupeds, at least at the higher level of abstraction of actions learned.
>
> ### Simplicity and Future Work
> Second, we agree with the reviewer regarding the simplicity and uniqueness of the task exhibited. This is, in fact, acknowledged in the **limitations section A2** of the appendix. Aspects that are left for future work include the inclusion of multiple behaviors that go beyond reaching landmarks (such as obstacle avoidance, direction following, etc.), which will go hand in hand with the inclusion of memory. This will enable **Flex** to serve as the base architecture for complex robot navigation tasks across multiple domains.
>
> ### Generalization and Controlled Experiments
> Nevertheless, the goal we set with this project is to identify the ingredients necessary for generalization in both environments and goal targets. We hope the reviewer will appreciate that a single controlled experiment—with ablations and analysis in terms of data collection, patch feature resolution, and policy network choice—provides a unified playground for clear comparisons, analysis, and conclusions.
>
> In this context, we do identify a solution that is able to achieve the specified task in the real world from simple simulation data, with open dictionary goals specified via text by a user. It is in that sense that our generalization claim holds.
>
> #### Reference
> - [1] "Drive Anywhere: Generalizable End-to-End Autonomous Driving with Multi-modal Foundation Models" by Tsun-Hsuan Wang, Alaa Maalouf, Wei Xiao, Yutong Ban, Alexander Amini, Guy Rosman, Sertac Karaman, and Daniela Rus.
>
> ## Weakness 2
>
> We thank the reviewer for this very insightful question, which sheds light on important differences between Vision-Language Models (VLMs) and their capabilities.
>
> ### Embedding vs. Fusion of Image and Text
> Both **CLIP** and **ImageBind** can embed pairs of images and text together, but not directly in the sense of fusing the two modalities into a single embedding. Instead, these models independently process images and text into separate embeddings that exist in a shared latent space. Thus, there is no direct interaction or integration between the modalities during the embedding process. Their embeddings are simply aligned in the shared space. This approach is well-suited for tasks like retrieval, similarity measurement, and zero-shot learning across modalities but does not produce a joint representation of paired inputs.
>
> ### Direct Fusion in BLIP-2
> On the other hand, **BLIP-2** integrates image and text inputs directly, allowing the model to process the interactions between the two. This fusion enables a deeper understanding and reasoning about the relationship between the image and text, which is essential for tasks requiring combined context, such as **robotic navigation from instructions**.
>
> ### Selection of Models
> We conscientiously avoid models that do not produce a unified fused representation from an image and text pair. Other models that offer such feature representations—like **LXMERT**, **ViLBERT**, and **UNITER**—share similar architectures. We have thus proceeded to select the most popular models in use to validate our approach.
>
> We are happy to discuss with the reviewer the possibility of extending our analysis to specific feature extraction models for the camera-ready version of the work.

---

> > ### Author Response · Authors · 2024-11-19
> > **Initial reply continued**
> >
> > ## Weakness 3
> >
> > Thank you for your insightful comments. We agree that clarifying the advantages of end-to-end learning in our setup is essential for demonstrating the impact of our work.
> >
> > ### Clarification of Goals
> > First, we want to emphasize that our goal is **not** to solve the visual navigation task itself, and we attempted to make this clear in the paper. Instead, our objective is to provide a **minimal approach** for using Vision-Language Model (VLM) features in robotic tasks to achieve **open-set generalization** with minimal training and data. We used navigation as an example to illustrate this approach.
> >
> > ### Challenges in Our Setup
> > In our setting, the challenges stem from the **minimality constraints**. Specifically, being trained on only **two objects** and needing to generalize to an open set, as well as being trained in a **single environment** while generalizing to new unseen ones. We showcase how **minimal design modifications** can lead to robust generalization in these scenarios.
> >
> > ### Advantages of End-to-End Learning
> > As mentioned by the reviewer, our approach offers an **end-to-end learning method** for utilizing VLM features in downstream tasks. Here, we try to clarify the advantages of our end-to-end approach over modular approaches:
> >
> > 1. **Adaptability**:
> >    It can serve as a **plug-and-play solution** for new tasks, requiring only changes to the loss function and training data. This flexibility enables easy adaptation to different tasks, as the process is not specific to the learned task.
> >
> > 2. **Minimality and Simplicity**:
> >    We use a minimal approach to leverage VLM features for robotic tasks, requiring a very small amount of training, data, and model adaptations.
> >
> > 3. **Reduced Error Accumulation**:
> >    Because it is an end-to-end approach, it reduces the risk of **accumulated errors** across various parts of the system, which can often occur in modular setups.
> >
> > ## Weakness 4
> >
> > As we are the first to integrate this **tiny-learning, data-short approach** with **open-set capabilities** into a robotic framework, we initiated our research with a simpler setting. Our goal was to establish a **foundational proof-of-concept** that demonstrates the feasibility of adapting VLM features in such a simple way to robot tasks, allowing **open-set text instruction** at the object and environment level.
> >
> > ### Focus on Core Challenges
> > Starting with **basic instructions** allowed us to focus on the **core challenges** of this novel integration without the additional complexities that come with handling more intricate language instructions. This approach enabled a deep understanding of all components, ensuring that the essential aspects of the framework were well-explored.
> >
> > ### A Critical First Step
> > We believe that this approach is a **critical first step** toward more sophisticated user-interactive systems with minimal model and data design and training. By successfully implementing and validating the fundamental aspects of **Flex**, we set the groundwork for future developments that can incorporate **advanced language comprehension capabilities**.
> >
> > Following your insightful comments, we have refined the writing to clarify these statements more clearly.

---

### Official Review · Reviewer_mJid · 2024-11-04

**Soundness:** 3
**Presentation:** 3
**Contribution:** 3
**Rating:** 5
**Confidence:** 3

**Summary:**

This paper introduces FLEX, a method for making drones understand and follow natural language commands. Instead of building everything from scratch, it cleverly uses an existing vision-language model (BLIP-2) to process both camera images and text instructions. The system was trained on simple simulated scenarios with just two colored balls, yet surprisingly, it worked well in real-world tests with various objects. The key innovation is how it breaks down camera images into patches and processes them alongside text commands, allowing the drone to understand both what it sees and what it's looking for.

**Strengths:**

1. Integrating visual navigation with the vision-language foundation model is an interesting direction.
2. The performance of this paper is reasonable with targeting and query on a limited set of objects.
3. The paper writing is in general good except the introduction.
4. The paper evaluates their methods in both simulation and real scenarios.

**Weaknesses:**

1. The paper writing is strange where the introduction section is quite similar to related work.
2. The training data seems to be rich with 1, 1M, 2, 2M, however those four datasets contain only a limited amount of target, and query, which I think is not enough for an open world navigation paper.
3. In addition the paper didn't handle how to solve image sequences for navigation work, which means that they predict action frame by frame.
4. One interesting thing is that in order to correctly navigate in the scene, actually we need to know the depth information, however, the amount of data the paper trained is not enough for that kinds of generalization.

**Questions:**

1. I am mostly interested on how the evaluation data is different from the training data. I am happy to fix my score, if I could understand the performance is not from scene memorization.

---

> ### Author Response · Authors · 2024-11-19
> **Initial reply**
>
> We sincerely thank the reviewer for their professional evaluation and detailed feedback. We look forward to further engaging during the open discussion period. To initiate this dialogue, we have provided comprehensive responses to all the issues and questions raised in your initial review, hoping it will encourage you to increase your score.
>
> ## Weakness 1
>
> We agree with the reviewer and thank them for contributing to an improvement in the presentation. In the revised version of the manuscript, we have moved some of the related work citations from the introduction to the related works section and merged the Introduction with the Preliminaries.
>
> Thanks for the careful reading.
>
> ## Weakness 2
>
> The reviewer makes an important and valid point. We provide some nuance as to its pertinence to our approach.
>
> First, we note that while the number of image-action pairs in our dataset is slightly large (as this is a regression task), the number of demonstrations is considerably small, totaling **300**. Furthermore, the training dataset includes only **1 or 2 distinct objects**, making the diversity of training data even more limited. Additionally, our training is conducted in just a **single environment**. Despite these constraints, we demonstrate **open-set generalization** across text, objects, and environments. Compared to previous works, this is the first study to achieve such robust generalization from a limited set of environments, objects, and demonstrations.
>
> Second, our argument would be that this is **not** an open-world navigation paper in its widely adopted sense. The emphasis in this work is not to train a robot navigation foundation model or to build one via an ingenious combination of vision and language models. The difference to such approaches is stated in our **Related Works** in **section 6** under “Contrasting recent approaches with our goal”.
>
> Indeed, our goal is stated in our **Preliminaries**, section 2, part of which is reproduced here for the reviewer’s convenience:
> > **Problem statement**: We seek to establish the bare design criteria for training robust, text-instructed, end-to-end control agents. Specifically, our goal is to delineate the conditions for effective leveraging of off-the-shelf models to extract meaningful features suitable for compact downstream policy networks.
>
> In other words, what learning representations are required for the emergence of object and environment generalization in end-to-end navigation? This is a very different research question from that of designing a comprehensive navigation model. Furthermore, in terms of **dataset design**, we try to answer the question (also in section 2):
> > **Dataset design**: What is the minimal degree of data diversity required to obtain sufficiently rich feature representations?
>
> ## Weakness 3
>
> The reviewer highlights a significant systems limitation we clearly state in our **Discussion**, **Appendix A.2**:
>
> > **Limitations and Future Work**: The framework presented in this manuscript is limited to instantaneous decisions. Indeed, the policy can only act with information from the current image and has no access to a history of representations or actions. We are keen to incorporate our potent multi-modal feature encoding scheme into sequential decision-making processes. This would enable **Flex** to go beyond generalization between environments and objects, and handle instructions over actions, sequences of steps, and behavior modes.
>
> As mentioned, this is left for future manuscripts, as we believe it to be a crucial capability that will enable a range of behavior required for the system to be useful for long-horizon applications.
>
> However, we maintain that this does not dilute the contribution of the paper, which we see as less of a novel systems design but rather an investigation of the relationship between **learning representations** and the emergence of **robust generalization** (albeit on a simple scenario).
>
> Indeed, we aim to devise a **minimal approach** for adapting the generalization capabilities of VLMs to downstream robotics tasks, with minimal training/data/model modification.

---

> ### Author Response · Authors · 2024-11-19
> **Initial reply continued**
>
> ## Weakness 4
>
> The reviewer brings up a good point that a more difficult formulation of a navigation task would be to navigate to a precise distance away from the target object.
>
> ### Clarification of Task Evaluation
> We would like to highlight to the reviewer that, as described in **Section 4.1**, the evaluation focuses on whether “the agent can navigate towards the user-instructed object and center it in the middle of the frame.” While this does not explicitly enforce navigation to a precise distance, it inherently involves assessing the agent’s ability to maintain an appropriate **relative positioning** with respect to the target. Moreover, we test the model across diverse conditions, including:
> - Out-of-distribution visual backgrounds
> - Target objects of varying sizes
> - Different instructions
> - Simulation-to-real transfer
>
> This demonstrates robustness in meeting this task specification.
>
> ### Data Type and Framework Consideration
> More importantly, we do not believe the question here is about the **amount of training data used**, but rather the **type and modalities of data** considered for a specific framework. In this work, we operate within the **visual (and visual-only) end-to-end control learning setup**. The rationale is that this is a popular approach applicable to a wide array of robotics platforms and applications, avoiding problem overspecification. It is also a framework that readily reaps the progress in **computer vision** and **multimodal foundation models** enabled by the huge amounts of image data available on the internet (which we explore in this work).
>
> ### Blueprint for Future Tasks
> Thus, we argue that although our selected task is rather simple and perhaps ill-specified in the sense of accurate distance monitoring, it serves as a **blueprint approach** that can be extended (albeit with further improvements and development) to other visual navigation tasks across various platforms (e.g., ground and sea vehicles, humanoids, quadrupeds, etc.) in a **sensor-agnostic fashion**.
>
>
> ## Question 1
>
> We thank the reviewer for these comments, and we believe they highlight an important point. Below, we provide a clear description of the experimental setup and results. We are happy to emphasize these details more prominently in the paper following the reviewer’s insightful suggestions.
>
> ### Environment
> Our policy model was trained in a single simulated environment called *Samurai*. At test time, the model was evaluated across three distinct scenarios:
> 1. Same Training Environment (*Samurai*)
> 2. A New Simulated Environment (*Stadium*)
> 3. The Real World
>    - Tests were conducted in various and diverse surroundings.
>
> ### Objects
> - **Training Data**:
>   - Training utilized only two spherical objects with specific colors: blue and red.
>   - These were the only objects present in the training data.
>
> - **Generalization Performance**:
>   The model demonstrated significant generalization to open-set scenarios:
>   1. Mixed Color Shapes (Simulation):
>      - Generalized to objects with varying shapes and colors, beyond those seen during training.
>   2. Open Dictionary (Simulation):
>      - Tested on a wide range of simulated objects, including:
>        - A blue sphere
>        - A light-colored Jeep
>        - An Australian cattle dog
>        - A brown horse
>        - A tall and narrow palm tree
>        - A toy space rocket
>        - A whole watermelon
>   3. Open Dictionary (Real World):
>      - Evaluated on various real-world objects during drone navigation, including:
>        - A yellow star
>        - A bicycle
>        - A man with a wig
>        - And more.
>
> We are happy to emphasize these points more prominently in the paper based on the reviewer’s valuable feedback.

---

### Official Review · Reviewer_2xnd · 2024-11-05

**Soundness:** 2
**Presentation:** 3
**Contribution:** 2
**Rating:** 5
**Confidence:** 3

**Summary:**

This work studied robust multimodal representations for drones. It combines spatial and lexical features via patch-wise descriptors from VLMs. The training pipeline for closed-loop visual navigation agents that generalize across unseen environments, using real-time natural language instructions to achieve adaptability well beyond the training scope. Extensive experiments on drone fly-to-target tasks, showcasing the ability in generalization and real-world deployment

**Strengths:**

1. This paper is well written and easy to read.
2. The experiment part is good with analysis in feature robustness quality. Real-world experiments are given.

**Weaknesses:**

1. The motivation of using mask-based patch features is unclear to me. I think it can be simply replaced by ViT style patch feature encoding and attention-based interaction. And for some downstream policy nets like MLP, the spatial feature is eliminated by average pooling. Why the spatial dimension is necessary?
2. Following above, I do not find any ablation studies on different feature designs.

**Questions:**

See weakness part.

---

> ### Author Response · Authors · 2024-11-19
> **Initial reply**
>
> We sincerely thank the reviewer for the thorough assessment and constructive feedback, which helped us to enhance our paper's clarity. We are confident that we have adequately addressed all of your concerns. If you have any further questions, we are eager to continue the discussion during the review period. Thank you!
>
> ## Reply to question 1
>
> We thank the reviewer for touching a crucial point of the work. We remind the reviewer that the focus of this work is the identification of minimal requirements to obtain generalization in navigation to new objects in new scenes (in simulation or in the real world). The natural assumption made is that foundation models are essential in bridging the generalization gap, especially for the handling of language-provided concepts outside the scope of the policy training dataset. With that said, the question becomes about the suitable usage of VLM models (here BLIP-2) to achieve this objective.
>
> Hence, the short answer to the first question is that it does not work. In fact, if we understand the reviewer correctly, the setup described corresponds to the case presented in Figure 3 with patch resolution 1. Here the entire image goes through the VLM which does use its underlying ViT 16x16 patch attention mechanism to encode the image. With an attention based policy trained on such features, the closed loop navigation performance is close to 0% on all tests whether in or out of distribution. The introduction of spatial masking is accompanied by first the emergence of success, and further the improvement of success as the resolution increases (again in Figure 3).
>
> Now to address the point regarding the MLP average pooling, we bring the reviewer’s attention to the fact that the MLP acts on spatially encoded patch features and does not interfere with the encoding process. Thus, no loss of spatial information occurs before the decision policy. The linear layers of the MLP pre-pooling operate on the spatially encoded patch features, learning the appropriate relationships and associations between spatial positions and actions. These operations are completed before average pooling, which serves only as a final aggregation step. Thus, the MLP policy still sustains reasonable levels of success in easy scenarios (Figure 4).
>
> In short: (i) The standard ViT model is not designed to map input images directly to policy outputs. As a result, the final descriptor output of a ViT does not capture the necessary information required for learning a generalizable policy for navigation or other downstream task. Additionally, (ii) The patch features produced by intermediate layers of the ViT before the output stage do not share an embedding space with the text data, as, they are not trained to represent patch semantics fused with text information. To this end, for text-based navigation tasks and many other downstream tasks spatial-text-patch fused features are required. To enable text-patch fusion embeddings, our novel approach is required. Finally (iii) The policy network leverages all the available information from our method, including semantics, text-fusion embeddings, and spatial data, to effectively learn from these diverse sources. So the main idea is to provide the policy network (which is treated as the decision maker) with all possible relevant features! And it will learn to make a decision.
>
> ## Reply to question 2
>
> We appreciate the reviewer's comment. As we think that this is a very important part of the paper, we would like to clarify it to the best we can. Our ablation study focused on three components that cover all layers of the framework:
>
> 1- Feature Extractor Model: We investigated the minimal approach to obtaining a feature extractor that provides all necessary features without additional training, thereby allowing good generalization due to its features. We compared our approach to standard Vision Transformer (ViT) features and found that they do not perform effectively without the high-resolution features. Regarding the use of interlayer patch features, this approach cannot generalize to unseen text data because the patch features are not trained to fuse text with image descriptions. In fact, we initially experimented with this method using CLIP patch features but were unable to generalize text commands to objects not seen in the training data.
>
> 2- Policy Network Choices: We experimented with various popular architectures for the policy network, including ViT, Convolutional Neural Networks (CNNs), and Multi-Layer Perceptrons (MLPs). Our ablation study shows slight advantages for ViT, as detailed in Section 5.3, and we are happy to test more if any suggestions are provided.
>
> 3- Training Data Importance: We strongly believe that data is crucial. Therefore, we focused on determining the sufficient amount of training data needed in terms of objects, text commands, and more.
>
> We want to emphasize that we are more than happy to consider and adopt any relevant suggestions we receive.

---

### Author Response · Authors · 2024-11-19
**General reply**

We sincerely thank the reviewers for their positive feedback and valuable constructive criticism. Your professional review and careful reading have already helped us improve our work. We have thoroughly addressed all the comments raised during the initial review. If further clarification is needed, please do not hesitate to reach out. Your engagement is highly appreciated. We will be adapting the manuscript as the discussion phase progresses and provide you with the revised version by the end of the period.

Our response was slightly delayed as we undertook thorough double-checking to ensure that our stated claims, especially regarding the use of other features, are indeed correct. We thank you for your patience.

---

### Author Response · Authors · 2024-11-21
**Looking forward to our discussion**

Dear Reviewers,

Thank you once again for taking the time to review our work and provide valuable feedback. Your insights have been instrumental in enhancing the quality of our paper. We hope that our clarifications regarding the scope, objectives, and uniqueness of the work have effectively addressed your concerns.

As the discussion period continues, we are happy to engage in further dialogue and welcome any additional questions or feedback you may have. Please don’t hesitate to reach out—we are more than willing to address them promptly.

The authors!

---

### Author Response · Authors · 2024-11-25
**Feedback as rebuttal nearing end**

Dear Reviewers, ACs, and SACs,

We value the constructive feedback provided by the initial reviews. However, we hope we can get some benefit from the openly communicated reviewing process and have some engagement from the reviewers to further improve the paper.

At the moment, we have not yet had the chance to engage with any of the reviewers after addressing their comments.
As the discussion period is coming to an end, we would like to note that we are here to address any lingering issues and comments, even at short notice. We believe that we have adequately addressed all the reviewers’ concerns and issues but remain available to provide clarifications and changes to the manuscript.

Thank you and we look forward to hearing back.

---

### Author Response · Authors · 2024-11-27
**Revision submitted**

We wish to inform the reviewers and the chairs that we have submitted a revised version of the manuscript taking into account the reviewers' comments namely (changes in purple in the submitted revision):

- Rewriting of the intro and related works section for improved presentation and clarity
- Clarification w.r.t weakness 4 from reviewer BS4x at the end of the intro
- A summary of the test setup to ensure generalization to new environments and objects is clearly stated (new subsection 4.3)
- Clarification regarding the failure of entire image encoding in providing useful features for robot action (section 5.2)

We thank the reviewers for their valuable input, and believe the manuscript changes have addressed their main concerns. With the PDF changes no longer supported, we remain available to answer any additional questions the reviewers might have.

---

### Meta-Review · Area_Chair_8h2q · 2024-12-23

**Metareview:**

This paper investigates the use of frozen pre-trained vision-language models (VLMs) with the philosophy of making it lightweight and minimal in terms of both training and data. The approach uses a patch-based representation along with a proposed masking approach to perform feature extraction. Results are demonstrated on a drone scenario tested in simulation and real world. Reviewers appreciated that the topic of VLM integration towards vision-based navigation is interesting and that the paper is easy to read. However, overall reviewers had significant concerns about the paper, including the motivation of the method (e.g. use of masking, 2xnd), the limited amount and complexity of evaluation tasks (BS4x, mJid) and situation with respect to other methods (e.g. GOAT mentioned by BS4x). and lack of comprehensiveness in terms of other feature extractors (BS4x). After reviewing the rebuttal, many of these issues still remain. The use of VLMs, both in terms of frozen models and fine-tuned, is a topic that is both interesting but also being explored by a large number of works (including some cited by the authors, methods like GOAT, etc.). Such methods are often compared across a number of vision-based navigation benchmarks with varying complexity, open-vocabulary variants, etc. to demonstrate the universality of the proposed architectures proposed. The current paper proposes another point in this space of possible designs (which is quite large), but only shows results on very limited settings. Further, aspects such as generalization (e.g. what variations, distribution shifts, new objects/scenes/tasks will it generalize to?) is not explored at all. As such, I agree with all of the reviewers that this paper is below the bar for acceptance. I recommend that the authors address the major concerns of the reviewers and significantly expand both the justification of the design and the scope of evidence for its effectiveness.

**Additional Comments On Reviewer Discussion:**

Many concerns were raised by the reviewers, including the lack of comprehensive evaluations that are common for the vision-based navigation literature. This latter weakness especially was not addressed by the rebuttals.

---

### Decision · Program_Chairs · 2025-01-22

Reject